# One Step In Situ Co-Crystallization of Dapsone and Polyethylene Glycols during Fluidized Bed Granulation

**DOI:** 10.3390/pharmaceutics15092330

**Published:** 2023-09-16

**Authors:** Shizhe Shao, David Bonner, Brendan Twamley, Abhishek Singh, Anne Marie Healy

**Affiliations:** 1School of Pharmacy and Pharmaceutical Sciences, Trinity College Dublin, D02 PN40 Dublin, Ireland; shaos@tcd.ie (S.S.); bonnerd@tcd.ie (D.B.); 2SSPC, The Science Foundation Ireland Research Centre for Pharmaceuticals, School of Pharmacy and Pharmaceutical Sciences, Trinity College Dublin, D02 PN40 Dublin, Ireland; 3School of Chemistry, Trinity College Dublin, D02 PN40 Dublin, Ireland; twamleyb@tcd.ie; 4Janssen Pharmaceutica NV, 2340 Beerse, Belgium; asing195@its.jnj.com

**Keywords:** polymer co-crystals, fluidized bed granulation, co-crystallization in situ

## Abstract

Several studies have demonstrated the feasibility of in situ co-crystallization in different pharmaceutical processes such as spray drying, hot melt extrusion, and fluidized bed granulation (FBG) to produce co-crystal-in-excipient formulations. However, no previous studies have examined such a one step in situ co-crystallization process for co-crystal formulations where the coformer is a polymer. In the current study, we explored the use of FBG to produce co-crystal granules of dapsone (DAP) and different molecular weight polyethylene glycols (PEGs). Solvent evaporation (SE) was proven to generate DAP-PEGs co-crystals at a particular weight ratio of 55:45 *w*/*w* between DAP and PEG, which was subsequently used in FBG, using microcrystalline cellulose and hydroxypropyl methyl cellulose as filler excipient and binder, respectively. FBG could generate co-crystals with higher purity than SE. Granules containing DAP-PEG 400 co-crystal could be prepared without any additional binder. DAP-PEG co-crystal granules produced by FBG demonstrated superior pharmaceutical properties, including flow properties and tableting properties, compared to DAP and DAP-PEG co-crystals prepared by SE. Overall, in situ co-crystallization via FBG can effectively produce API-polymer co-crystals and enhance the pharmaceutical properties.

## 1. Introduction

Today, oral delivery is still one of the most popular choices for drug delivery due to high patient acceptability and compliance compared to other delivery routes. According to a recent study, oral solid dosage (OSD) forms of pharmaceuticals comprise around 90% of the global pharmaceutical product market, and 60% of small molecule active pharmaceutical ingredients (APIs) are designed into OSD formulations [1]. However, designing suitable OSD formulations is still challenging because the absorption of APIs from the gastrointestinal (GI) tract is a complex process and will be impacted by the API’s properties such as dissolution rate, solubility, permeability, and physicochemical stability [2,3,4,5].

The Biopharmaceutics Classification System (BCS) is a well-established system to evaluate APIs’ properties for OSD formulation design. The BCS involves categorization of APIs into four classifications according to their solubility and intestinal permeability. Both properties are the key factors affecting drug absorption. Approximately 40% of the drugs on the market and 90% of drugs in the development phase are troubled by their low solubility [6].

Co-crystals constitute a well-studied solid form that shows the potential to improve various pharmaceutical properties of APIs, in particular solubility and dissolution rate [7,8]. The Food and Drug Administration (FDA) has provided a definition for co-crystals since 2018, whereby, co-crystals are crystalline materials composed of two or more different molecules, typically API and co-crystal formers (“coformers”), in the same crystal lattice [9]. The crystal lattice of co-crystals is normally composed of two or more different small (low molecular weight) molecules. The co-crystal coformer is either a small molecule excipient, normally from the Generally Recognized as Safe (GRAS) list or another small molecule API (a drug–drug co-crystal) [10,11]. Compared to the well-studied API–small molecule co-crystals, the study of API–polymer co-crystals is still an area that has not been fully explored and exploited. Moreover, polymeric crystalline inclusion complexes (CICs) are liable to be confused with API–polymer co-crystals. The clear differences between these two solid forms have been well explained by Chappa et al. Firstly, a characteristic of co-crystals is strong hydrogen bonding between the different molecules, but in CICs, there is a polymer inclusion in the API’s crystal channel, with weak or no hydrogen bond interaction. Secondly, a single unique endothermic melting behavior can be observed for co-crystals, but a dissociation event after API melting can be observed for CICs [12,13,14]. Compared to well-studied small molecule co-crystals, only a few cases of polymer co-crystals have been identified and characterized, but the APIs concerned are not delivered orally [15,16,17,18]. However, the more recent discovery of dapsone–polyethylene glycol (DAP-PEG) co-crystals has highlighted the emerging subset of polymer co-crystals in pharmaceutical co-crystals [14], which brings much more opportunities and options for pharmaceutical formulation scientists to enhance an API’s physicochemical properties. Dapsone is a BCS class II API and DAP-PEG600 co-crystal was shown to have a higher solubility than DAP in different physiological pH conditions [14]. 

Aside from discovering novel co-crystals, challenges of how to scale up the production of co-crystals and how to develop a uniform, stable, and well-behaved crystalline solid dispersions (CSDs) of co-crystals in a one step in situ co-crystallization process are gaining increasing attention from formulation scientists. The normal approaches used for generating or discovering co-crystals at small scale include neat milling [19,20], liquid-assisted milling [21], and solvent evaporation (SE) [22], while scale-up processing techniques for in situ co-crystallization, such as spray drying, hot melt extrusion, and fluidized bed processing, are gaining popularity. All three of these scale-up techniques have been proven to generate pure co-crystals either alone or together with suitable excipients [23,24,25,26]. Furthermore, there is already a growing number of studies that investigate how different excipients in such one-step production processes affect the co-crystal formulation properties, such as crystallinity, dissolution, rate of dissociation, tabletability, etc. [26,27,28,29]. Nevertheless, to the best of our knowledge, there are no studies to date that investigate the production of polymeric co-crystals in such one step in situ co-crystallization processes.

In this study, we focused on the previously reported DAP-PEGs co-crystals [14] as examples of polymeric co-crystals to assess if formulations of polymeric co-crystals can be generated in a one step in situ co-crystallization process employing fluidized bed granulation (FBG). The objectives of this study were (1) to determine whether formulations containing polymer co-crystals can be obtained by FBG, (2) to investigate the possibility of the polymer co-former of such co-crystals acting as a binding agent, and (3) to assess whether the tablets made from co-crystal granules produced by FBG have better tabletability than equivalent physical mixtures (PMs) and if the dissolution performance of tablets prepared from such granules is better than equivalent PMs.

## 2. Materials and Methods

### 2.1. Materials

Dapsone (DAP form III) was purchased from Glentham Life Sciences (Corsham, UK). PEG 400, PEG 4000, and PEG 6000 were purchased from Tokyo Chemicals Incorporated (Zwijndrecht, Belgium). PEG 600, hydrochloric acid (HCl), and acetone were purchased from Sigma Aldrich (Arklow, Ireland). PEG 1000, PEG 1500, and methanol were purchased from Thermo Fisher Scientific (Dublin, Ireland). Microcrystalline cellulose (MCC) (Avicel PH 102) was a gift from DuPont (Cork, Ireland). Hydroxypropyl methylcellulose (HPMC) (27–30% methoxy content and 7–12% hydroxypropoxy content, 40–60 cp, around 22,000 g/mol)) was purchased from Alfa Aesar (Ward Hill, MA, USA).

### 2.2. Preparation Methods

#### 2.2.1. Solvent Evaporation (SE)

First, 1 g of 55.0:45.0 *w*/*w* and 57.5:42.5 *w/w* of DAP/PEG (each of PEG 400, 600, 1000, 1500, 4000, and 6000) (Table 1) were dissolved in 100 mL of a 1:1 *v/v* acetone and methanol solvent mixture and stirred and heated to 50 °C until complete dissolution had occurred. The resulting solutions were left to evaporate in a fume hood overnight.

#### 2.2.2. Fluidized Bed Granulation

A fluidized bed system (Mini Glatt with Micro-kit, Glatt, Binzen, Germany) was used for a 50 g scale study. The bottom spraying mode was used in all cases. MCC was used as a filler and added into the fluidized bed as a powder. HPMC was used as a binder and was co-dissolved with the DAP and PEG (55.0:45.0 *w/w*) in a 1:1 *v/v* acetone and methanol solvent mixture. After the full dissolution of DAP, PEG, and HPMC, the solvent was sprayed into the Mini Glatt tower containing the MCC powder. Fluidized bed granulation (FBG) process variables and formulation compositions are shown in Table 2 and Table 3, respectively.

### 2.3. Material Characterization

#### 2.3.1. Powder X-ray Diffraction (PXRD)

PXRD was carried out in reflection mode using a benchtop X-ray diffraction instrument Rigaku Miniflex II (Rigaku, Tokyo, Japan) equipped with a CuKα x-ray source and Haskris cooling unit (Grove Village, IL, USA). The samples were front-loaded by gently pressing using a glass slide on a zero-background silicon sample holder. The PXRD patterns were recorded for 2θ ranging from 5° to 40° with a step scan rate of 0.05° per second.

#### 2.3.2. Modulated Differential Scanning Calorimetry (mDSC)

Samples were analyzed using a Q200 MDSC (TA Instruments, Leatherhead, UK) equipped with a refrigerated cooling system (RCS-90) and purged with inert dry nitrogen gas at a flow rate of 50 mL/min. Indium standard was used for temperature and melting enthalpy calibration and validation. Around 2–5 mg of each sample was weighed into a TA Instruments standard aluminum pan which was sealed with a lid which was then pierced with two pinholes. The sample pan was first equilibrated to −40 °C and maintained for 2 min before being heated to 200 °C at 5 °C/min heating rate. The equipment was set to modulation mode, with 0.8 °C of modulation every 60 s. The measurement of the glass transition temperature (Tg) for HPMC was performed using the steps described above, with an additional heating process to reach 100 °C at a heating rate of 5 °C/min. The sample was then kept at 100 °C for 10 min before being cooled to −40 °C at a ramping rate of 5 °C/min, prior to the analysis. The data were acquired and analyzed using Thermal Advantage and Universal Analysis software (version 4.5A, TA Instruments, Leatherhead, UK), respectively. The analysis was performed in triplicate.

#### 2.3.3. Thermogravimetric Analysis (TGA)

Thermogravimetric analysis (TGA) was performed using a QA-50 device (TA instruments, Elstree, UK). Open aluminum pans containing 2–10 mg of the sample were heated at a rate of 10 °C/min under an inert nitrogen atmosphere from ambient (approx. 20 °C) to 200 °C. Weight changes observed to 120 °C enabled calculation of the residual solvent content (RSC) by measuring the % weight loss of the sample.

#### 2.3.4. Particle Size Analysis

Particle size distribution analysis was undertaken by the dry method in a laser diffraction particle size analyzer (Mastersizer 3000, Malvern Instruments Ltd., Worcestershire, UK) equipped with an Aero S dry powder disperser unit with 2 bar pressure (or 3 bar, where indicated) and feed rate of 50%. Malvern Mastersizer 3000 software (version 3.50) was used to determine the d_10_, d_50_, and d_90_ values, which are the particle sizes corresponding to the 10%, 50%, and 90% points of the cumulative volume distribution curve, respectively. The d_50_ is the median particle size of the volume distribution. The width of the size distribution was expressed by the span value, calculated from the d_10_, d_50_, and d_90_ values, using Equation (1).
Span = (d_90_ − d_10_)/d_50_(1)

#### 2.3.5. Fourier Transform Infrared Spectroscopy

Infrared spectra were recorded on a PerkinElmer Spectrum 1 FT-IR Spectrometer (Waltham, MA, USA) equipped with a UATR and a ZnSe crystal accessory. Each spectrum was scanned in the range of 650–4000 cm^−1^. Data were evaluated using Spectrum v 5.0.1. software. Six scans of each sample were taken.

#### 2.3.6. Raman Spectroscopy

The spectra were collected by a PhAT probe connected to Kaiser RamanRxn spectrometer (Kaiser Optical Systems Inc., Ann Arbor, MI, USA), with a nominal laser beam diameter at the focal position of 6 mm. The probe was perpendicular to the horizontal tabletop and was a distance away from the tabletop. The Raman signal from the sample was collected by placing the sample directly underneath the probe. The exposure time was set to 2 s with 6 scans, using a laser power of 400 mW, over a range of 150–1850 cm^−1^. Verification of the Raman spectroscopy was checked regularly using cyclohexane.

#### 2.3.7. Scanning Electron Microscopy (SEM)

Surface images of the samples were captured by SEM using a Zeiss Supra Variable Pressure Field Emission Scanning Electron Microscope (Zeiss, Oberkochen, Germany), which was equipped with a secondary electron detector. Samples were applied using glue onto carbon tabs, mounted onto aluminum pin stubs, and sputter-coated with gold under a vacuum. The samples were analyzed at 2 kV.

#### 2.3.8. Imaging of Granules/Powder in Bulk

A USB microscope 9 MP (Conrad Electronic, Hirschau, Germany) was utilized to take images of granules/powder using the eScope software programme (version 1.1.7.17) (OiTez, Hongkong, China).

#### 2.3.9. Simulated PXRD Patterns

Simulated PXRD patterns were calculated from single crystal crystallographic data using Mercury software version 3.10.3 (Cambridge Crystallographic Data Centre, Cambridge, UK).

#### 2.3.10. Ultraviolet Spectroscopy (UV)

A Shimadzu UV-1700 absorption spectrometer (Tokyo, Japan) and QS 10,000 quartz cuvettes were used to build a DAP absorption calibration curve according to a previously published method [30]. Two 6-point calibration curves were built in the range of 1–10 µg/mL DAP in 60:40 *v/v* methanol/water and 60:40 *v/v* methanol/0.1 M HCl. A wavelength of 290 nm was used for both calibration curves.

To determine the DAP loading of the granules, 100 mg of granules was dissolved and diluted to a suitable concentration (1–10 µg/mL) in 60:40 *v/v* methanol/water. All experiments were carried out in triplicate.

### 2.4. Micromeritic and Tableting Properties Characterization

#### 2.4.1. Flowability Study of Granules/Powder

First, 1.5 g of powder/granules were poured into a pre-weighed 10 mL graduated cylinder. The bulk volume was recorded. After 500 mechanical taps to the 10 mL graduated cylinder in a Copley SVM12 (Copley Scientific Limited, Nottingham, UK) density tester, the initial tapped volume was noted before another 500 taps was conducted. If the volume did not change again after the second lot of 500 taps, the tapped volume was recorded. A further 500 taps were performed until the volume did not change, and the tapped volume was noted. The Hausner ratio was calculated to evaluate the flowability of the granules/powder using Equation (2), where ρ_B_ is the bulk density and ρ_T_ is the tapped density, with density calculated from the weight divided by the volume (bulk or tapped).
Hausner Ratio = ρ_T_/ρ_B_(2)

#### 2.4.2. Tableting Study

Tableting studies were carried out to compare tableting properties of pure DAP powder, DAP-PEG co-crystals prepared by SE (DAP-PEG SECCAs), and DAP-PEG co-crystal granules. Sufficient material equivalent to a 100 mg DAP was weighed out for each sample. Two tablets were prepared at 100, 150, and 200 MPa, respectively, using a Natoli NP-RD10 (Saint Charles, MO, USA) laboratory-scale single punch tablet press supplied with an Enerpac (Menomonee Falls, WI, USA) P-392 manual pump with an RC-104 hydraulic cylinder working in the range from 0 to 10 kN and standard 8-mm diameter flat punch and die tooling (I Holland Limited, Nottingham, UK). The pressure was released immediately after the desired compression pressure was reached. The ejection force was recorded on the tablets being pushed out from the die. The diameter of the tablet was measured using digital vernier calipers accurate to within 0.1 mm. The breaking force of the tablets was measured by using a Electrolab Tablet Breaking Force Tester (Lab unlimited, Dublin, Ireland). Tensile strength (MPa) was then calculated from Equation (3), where σ is the tensile strength (MPa), F is the breaking force (N), D is the tablet diameter (mm), and H is the tablet thickness (mm).
σ = 2F/(πDH)(3)

#### 2.4.3. Dissolution Study

A 708-DS dissolution apparatus (Agilent Technologies, Cork, Ireland) and apparatus I (basket apparatus) (Agilent Technologies, Cork, Ireland) were used for the dissolution studies. Tablets (equivalent to 100 mg DAP) were made from F5 (Table 3) and its equivalent physical mix (PM). Tablets were prepared at 200 MPa as detailed in Section 2.4.2. Dissolution testing was carried out following the method outline in the British Pharmacopoeia for dapsone tablets [31], whereby the tablets were placed into apparatus I baskets that were submerged into 900 mL of 0.1 M HCL at 37 °C and rotated at 100 rpm. Next, 5 mL samples were taken out from the dissolution medium at 5 min, 10 min, 20 min, 30 min, 45 min, 60 min, 90 min, and 120 min, and filtered through a 0.45 µm PTEE filter (Fisherbrand, Waltham, MA, USA). Methanol was added to all filtered samples to achieve a 60:40 *v/v* methanol/0.1 M HCl solution, which was subsequently diluted using 60:40 *v/v* methanol/0.1 M HCL solution to a concentration of 1–10 µg/mL for UV analysis. All dissolution studies were carried out in triplicate.

## 3. Results and Discussions

### 3.1. DAP-PEG Co-Crystals Generated by Solvent Evaporation

The first step in this study was to determine which weight fraction of DAP and PEG could produce physically pure co-crystals by SE. Subsequently, the same weight fraction of DAP and PEG would be used for the FBG study.

The SE method undertaken in this study is similar to that reported by Chappa et al., with the same solvent composition, but at a larger scale (1 g in 100 mL 1:1 *v/v* acetone/methanol solvent). Nevertheless, the polymer co-crystals are not like conventional co-crystals that usually can be defined as having a 2:1, 1:1, or 1:2 molar ratio between API and coformer [10], which means the determination of suitable weight fraction for co-crystal generation between API and polymer is challenging. Chappa et al. concluded that there are two values of weight fractions of DAP that can be used to generate pure DAP-PEG co-crystals, these being 55.0 and 57.3/57.6 wt.% (for replicate experiments) DAP. The first value of 55.0 wt.% was determined based on the fact that a unique single melting behavior with highest heat of fusion was observed (for DAP-PEG 6K and DAP-PEG 1450) when 55.0 wt.% DAP was used compared to other weight fractions (Appendix A). The second value of 57.5 wt.% (determined for DAP-PEG 6K) or 57.3/57.6% *w/w* (determined for DAP-PEG 1450) (which corresponds to an approximately 4:1 ratio of ethylene oxide monomer: DAP in the co-crystal was obtained by considering the heat of fusion of the DAP-PEG 6K and DAP-PEG 1450 co-crystals against the DAP weight fraction (% *w/w*) (Tamman’s triangle construction, Appendix A). The intersection of the linear fits can be considered as representing the precise stoichiometry of the co-crystal. From Chappa et al.’s calculations, 57.3–57.6 wt.% DAP should give the highest heat of fusion (and purest co-crystal forms), but there is a lack of experimental data provided (aside from the calculations based on Tamman’s triangle construction from the heats of fusion) to indicate if 57.3–57.6 wt.% could generate purer co-crystals compared to 55.0 wt.%. Hence, two weight fractions (55.0 and 57.5 wt.% DAP) were used in our SE experiments (Table 1) to determine which weight fraction gives the highest purity of co-crystals.

#### 3.1.1. XRD and DSC Analysis of SECC400 and SECC600

PXRD diffractograms of all raw materials are shown in Appendix A, where PEG 400 and PEG 600 are viscous liquids at ambient temperature, showing amorphous wide halos, and PEG 1000–6000 are semi-crystalline materials showing characteristic peaks at 19.3° and 23.5° 2θ [32]. There are a total of 6 polymorphs of DAP reported previously, 5 anhydrates (form I to form V), and 1 hydrate [33]. DAP form III is the most stable form and is generally used commercially [33]. The PXRD pattern of DAP raw material is characteristic of DAP form III, showing characteristic peaks at 11.6° and 13.1° 2θ [33] (Figure 1I).

The previous study by Chappa et al. demonstrated that all co-crystals formed between different molecular weight PEGs and DAP showed similar PXRD patterns, indicating structural uniformity, and the series of co-crystals could be identified as an example of an isomorphous crystal [14]. DAP-PEG 600 single crystal was generated by the vapor diffusion crystallization method, so its crystallographic data were used to determine if the SE samples had any residual/excess DAP or PEG [14,15].

Unfortunately, our results demonstrated that SECC400A, SECC400B, SECC600A, and SECC600B all contained pure starting materials left over. It can be seen from the diffractograms (Figure 1I,II) that a small single unique peak at around 7° 2θ observed in SECC400A and SECC600A represents DAP form II, and double unique peaks at 7.0–7.3° 2θ in SECC400B and SECC600B represent DAP form V [33]. SECC400A, SECC400B, SECC600A, and SECC600B all partially line up with the simulated co-crystal PXRD patterns with different peak intensities, which indicates the presence of co-crystals, but the intensities were affected by the impurities/non-co-crystallized materials present and/or the preferred orientation of the crystalline materials.

From the melting behavior (Figure 1III,IV and Table 4), DAP shows a minor endothermic peak at 81.4 ± 0.2 °C followed by a large endothermic event at 177.3 ± 0.2 °C. The first minor endothermic peak indicates DAP form III transforming to DAP form II and the second endothermic peak shows the melting behavior of DAP form II [25,33]. Melting of SECC400A and SECC400B was observed at around 75 °C, while melting of SECC600A and SECC600B was observed at around 99.5 °C [25,33], indicating the formation of co-crystals. The melting peak of SECC600A and SECC600B is consistent with the melting temperature of 100.7 °C previously reported by Chappa et al. [14]. The previous study did not investigate the possible co-crystal formation between DAP and PEG 400. Because the lower the PEG molecular weight the lower the melting temperature, a melting temperature around 75 °C of DAP-PEG 400 co-crystal, as was observed (Figure 1III and Table 4), seems reasonable. The small endothermic event in SECC400B and SECC600B at around 150 °C (Table 4) has been previously reported to be related to the transformation of DAP form V to form II [33]. This indicates that there was DAP form V leftover in SECC400B and SECC600B, which is consistent with our PXRD observations. Additionally, a small PEG endothermic melting event at −9 °C and 10 °C (Table 4) can also be observed in SECC400A and SECC600A/B, respectively. Thus, apart from co-crystal melting, the observed other endothermic events indicate that the SE approach cannot produce pure DAP-PEG 400 or DAP-PEG 600 co-crystals, regardless of the weight fraction of DAP used.

#### 3.1.2. XRD and DSC Analysis of SECC1000, SECC1500, SECC4000, and SECC6000

Unlike SECC400 and SECC600, SECC1000, SECC1500, SECC4000, and SECC6000 (at both weight ratios trialed) give identical PXRD patterns to simulated PXRD patterns of DAP-PEG 600 co-crystal (Figure 2), indicating the formation of co-crystals. In addition, there are no XRD characteristic peaks representing DAP polymorphs or PEGs observed in the samples produced from SE, also illustrating that there might be fewer impurities/non-co-crystallized materials in these SE solids compared to SECC400 and SECC600.

From DSC data of SECC1000-6000 (A and B, i.e., both weight ratios) (Figure 3 and Table 5), a single unique sharp co-crystal melting peak was observed for all solid samples. This phenomenon also indicates the successful formation of co-crystals. However, other endothermic events representing parent compounds (Table 5) can still be observed in SECC1500A and SECC4000A (pure PEGs), and SECC4000B and SECC6000B, indicating that these samples contain a small amount of impurity (DAP form V remains). Additionally, the enthalpy of fusion of SECC1500B, SECC4000B, and SECC6000B was clearly lower than SECC1500A, SECC4000A, and SECC6000A, suggesting that imperfect co-crystals were formed at 57.5 wt.% DAP weight fraction. 

#### 3.1.3. Determination of DAP Weight Fraction in DAP-PEGs Co-Crystals 

From the discussions in Section 3.1.1 and Section 3.1.2, despite the impurity(ies)/non-co-crystallized materials that can be observed in most samples, no matter which weight fraction DAP is used (55.0 wt.% or 57.5 wt.%), a single unique co-crystal melting behavior and comparable XRD patterns to the simulated PXRD pattern of DAP-PEG 600 co-crystal can always be observed. However, the co-crystals produced from the 55.0 wt.% DAP weight fraction always have a higher co-crystal heat of fusion compared to co-crystals produced from 57.5 wt.%, which indicates that the co-crystal generated at 55.0 wt.% DAP weight fraction may be physically purer or have a stronger intermolecular interaction between DAP and PEG than when 57.5 wt.% DAP is used. Thus, a 55.0 wt.% weight fraction of DAP was used for all FBG studies.

### 3.2. Solid-State Characterization Comparing SE Powder with FBG Granules

In this study, granules could not be successfully prepared by FBG using DAP on its own, but four co-crystal granule formulations (F1, F5, F9, and F10) were successfully prepared by FBG, where F1 does not contain any added HPMC binder.

To achieve a higher purity of co-crystals, DAP and PEGs with/without a binder were dissolved in the same solvent composition as was used in the SE study and the solution was sprayed onto the MCC powder [25]. Two FBG process conditions (Table 2) were used for granule production, whereby P1 and P2 were employed with a view to generating different droplet sizes and building different levels of “wetness” in the fluidized bed. It was anticipated that P1, with lower temperature and higher liquid feed rate, should generate a larger droplet size and create a wetter environment compared to P2 [34,35]. Moreover, the concentration of solid/solute was kept as high as possible for all the formulations (Table 3, except for F6), because a high concentration should cause a much higher binding affinity and should increase the chance of successfully obtaining granules [36]. This may be observed by comparing the outcome of F5 with that of F6 (Table 3), where F6 with a lower concentration did not result in any granules. Nevertheless, the failure to produce granules in the case of F7 and F8 might demonstrate that there is a concentration “sweet spot” where recovery of granules is possible. From the FBG study without any binders (Table 3, F1–F3), only F1 with PEG 400 and P1 could generate granules (Table 6). The reason why F2 and F3 failed but only F1 succeeded might be because of F1’s wetter powder bed and larger droplet sizes [37], and the higher affinity to filler provided by DAP-PEG 400 co-crystal compared to DAP-PEG 600 co-crystal. Note, the formation of co-crystals will be discussed later.

#### 3.2.1. XRD and DSC Analysis

PXRD patterns of all granules share the same peak positions as simulated PXRD of DAP-PEG 600 co-crystal (Figure 4), and co-crystals in FBG granules show an identical melting behavior to co-crystals in SE powders (Figure 5 and Table 6). Notwithstanding the wide halo beneath the crystalline Bragg peaks representing the amorphous MCC and HPMC, it might also indicate the existence of amorphous DAP-PEG or amorphous DAP. There are no glass transition events in the reversing heat flow (Appendix A) or recrystallization exothermic events (Appendix A) in the non-reversing heat flow, which indicates the crystalline nature of the co-crystal. Additionally, the glass transition of the HPMC, which occurred at a higher temperature (Tg = 162.42 ± 0.97 °C, Appendix A) than cocrystal melting, could not be observed in the reversing heat flow of the granules, probably because of the low composition of HPMC in the granules or because HPMC dissolved in the melted cocrystal, masking the glass transition process. Thus, despite the fact that FBG is a fast-evaporation method which can result in the generation of amorphous solid dispersions, both XRD and DSC results demonstrate that crystalline co-crystal rather than amorphous DAP-PEG/amorphous DAP was obtained from FBG. Although an amorphous solid can further enhance the solubility of the API, its physical instability issues cannot be ignored and require extra care and consideration [38,39,40]. Furthermore, from the DSC thermograms of FBG granules (Figure 5), no additional endothermic events (PEG melting or DAP melting, or DAP polymorphic transformation) were observed. Hence, from what has been discussed above, the FBG method could generate pure crystalline DAP-PEG co-crystals, in contrast to the SE method, for which almost all samples contained impurities (starting materials).

#### 3.2.2. TGA Analysis

All the co-crystal granules recovered from FBG have a residual solvent content (RSC) < 2%) (Table 7). The RSC of F1 (1.7 ± 0.1%) is clearly higher than that of F5 (1.0 ± 0.1%), F9 (1.1 ± 0.1%), and F10 (1.2 ± 0.1%). This is because F1 was produced using a different set of FBG processing conditions to the other three granule formulations, where a wetter fluid bed not only assists the granulation process (Table 3, F1–F3) but also increases the RSC of the granules. 

#### 3.2.3. Raman Spectroscopy

Raman spectroscopic analysis is a quick and non-destructive spectroscopic method. Co-crystal formation results in the formation of supramolecular synthons, with hydrogen bonding between different molecules existing in a shared crystal lattice [41,42]. Raman has been commonly utilized to verify co-crystal production [43] from pure single components or real-time monitoring of the co-crystallization process [44,45]. In terms of DAP-PEG co-crystals, the intermolecular hydrogen bond is formed between the primary amine group in DAP and the carbonyl group in PEG 600 (Appendix A), so a Raman spectral difference between DAP/PEG and co-crystals is expected.

Raman spectra of PEGs are shown in Appendix A. PEG 400, because of its liquid state at ambient temperature, shows wider peaks, but the same Raman shifts as other PEGs. Raman spectra of all SE powders have comparable peak positions to DAP and PEG (Figure 6I), whereby only some minor peak shifts can be observed, indicating the successful formation of co-crystals (Table 8). Moreover, it is difficult to say, based on the spectra, if there are any pure single components left over in SE powders as large spectra changes might not be expected even if there are changes in intermolecular interactions (transformation from single components to co-crystals) [46]. However, the spectrum of SECC1500B does not show the same Raman pattern as the other SECCs (Figure 6), as a peak at 1020 cm^−1^ is shown that does not represent the presence of DAP form III or PEGs or DAP-PEG co-crystals. This indicates that the co-crystallization of SECC1500B was imperfect, which is also shown by comparing the heat of fusion of SECC1500A and SECC1500B (Table 5). From what has been discussed above, after co-crystallization of DAP and PEGs, co-crystals were successfully generated, whereby a hydrogen bond connects DAP and PEG (Appendix A), resulting in minor Raman peak shifts. 

Raman spectra of granules show identical patterns to co-crystals generated by SE indicating the formation of co-crystals (Figure 6IV). In addition, no additional interactions between co-crystals and MCC/HPMC can be observed from the Raman spectra.

#### 3.2.4. FTIR Spectroscopy

Compared to Raman spectroscopy, which is excellent at revealing symmetric vibrations of non-polar chemical groups, FTIR spectroscopy is capable of revealing polar asymmetric vibrations. Both techniques in terms of co-crystal analysis are complementary to each other and have been commonly used together [49].

The FTIR patterns of SECCAs and SECCBs are the same as each other and comparable to pure DAP and PEG 6000 (Figure 7). The FTIR peak shifts are mainly observed in the peak attributed to the primary amine group in DAP, which is the group that is bonded by intermolecular hydrogen bonding to the PEG (Table 9 and Appendix A), indicating co-crystal formation in SECCAs, SECCBs, and granules, which is consistent with the DSC, XRD, and Raman analysis. Co-crystal formation from pure single components normally causes a 10 cm^−1^ to 15 cm^−1^ peak shift [50], which could also be observed in this study (Table 9) with one exception, whereby a 41 cm^−1^ peak shift from 3396 cm^−1^ to 3437 cm^−1^ is observed in all co-crystal systems. However, FTIR, like Raman, is still unable to determine the existence of the starting materials in SECCAs and SECCBs due to the overlayed peaks between starting materials and co-crystals, and the small amount of starting materials leftover. 

### 3.3. Pharmaceutical Properties of FBG Granules

#### 3.3.1. Granule Drug Loading

DAP loading of granules was tested by UV as indicated in Section 2.3.10. Because there was no excess DAP or PEG observed in the granules by PXRD, DSC, Raman, or FTIR analysis (Section 3.2), the co-crystal loading was calculated based on the weight fraction between DAP and PEG (55.0:45.0 *w/w*). If all of the co-crystals were bound to all of the filler, it could be expected that the co-crystal loading in the granules would be 30%. However, the actual co-crystal loading of different DAP-PEGs co-crystals ranges between 40% and 60% (Table 10), because some filler powder stayed trapped on the filter located on the top of the bed during the process, resulting in less filler to be bound. Furthermore, a trend in the results can be seen whereby the greater the PEG molecular weight, the greater the DAP and co-crystal loading in the granules. Only F1 has a co-crystal loading below 50%, but for the other three formulations, a co-crystal loading between 50% and 60% was achieved, probably because of the poorer binding ability of the DAP-PEG 400 compared to other formulations, which utilized HPMC as a binder. Furthermore, in the case of F1, more humid conditions are likely to arise inside the fluidized bed compared to the conditions utilized for the other three formulations (Table 7). It may be possible to further enhance the drug loading by adjusting FBG input settings based on a design of experiment approach, or by operating at larger scale, which may be worth considering in a future study.

#### 3.3.2. Particle Size Analysis

In this study, laser diffraction was utilized to analyze the particle size distribution of fluidized bed processed products to determine if granules are formed and their particle sizes. The particle size distribution of granules and DAP and MCC raw materials is shown in Figure 8 and Table 11. The particle size distribution of DAP is bimodal (Figure 8), indicative of lack-of-dispersion at 2 bar pressure, which was tackled by using a higher pressure of 3 bar in the laser diffraction particle size analyzer (Appendix A, Table 11).

The d_10_, d_50_, and d_90_ of the granule formulations were all greater than that of MCC. This signifies that the particle size enlargement/granulation process works for all molecular weights of PEG. It was surprising to see that F1, which is the “no-binder-involved” study, also had a larger particle size distribution than MCC, indicating that the DAP-PEG400 co-crystal itself can act as a binder for granule production. Given the fact that successful FBG runs typically require the presence of a binding agent such as HPMC to work effectively [51], the success of the “no-binder involved” study signifies the additional application of polymer co-crystals for enhancing API processability and formulation.

It was also noted that the larger the PEG molecular weight, the greater the particle size. It is also evident that while the F1 worked without the aid of a binder, there was a significant increase in particle size between F1 and F5, which can most likely be attributed to the presence of HPMC.

#### 3.3.3. Imaging of Granules/Powder in Bulk

Optical microscopy imaging demonstrates that DAP is a fine powder, with some agglomeration evident in the photomicrograph (Figure 9). On the other hand, in terms of F1, F5, F9, and F10 (Figure 9), the particles are granular in nature, with a larger particle size and uniform appearance.

#### 3.3.4. Morphology Study on Powder/Granules by SEM

Particle morphology of MCC, DAP, and granules was investigated by SEM (Figure 10). MCC and DAP have a small particle size with a non-regular particle shape (Figure 10). For all granules recovered by FBG (Figure 10), aggregates composed of small particles rather than coated layers of existing MCC single particles can be seen, indicating the success of granules formation rather than size-enlarged powders by FBG.

#### 3.3.5. Flowability of Powder/Granules

The flowability of the powder/granules is a key factor in pharmaceutical manufacturing processes [52]. Poor powder flow properties will result in challenging handling of the powder and will affect the tablet compression and capsule filling [53]. Fine powder might cause agglomeration and segregation [54], and DAP is an example of a fine powder leading to agglomeration (Figure 9 and Table 12). Granulation is normally applied to enhance the flow properties of API powders by enlarging the particle size of the fine powder [55]. In our study, all granules (Table 12) have an identical Hausner ratio of around 1.10 and have better flow characteristics than DAP raw material powder. The RSC in the granules seems to not affect the flow properties because the RSC of all granules (Table 8) is low RSC (<2%) and appears not to adversely affect flow.

#### 3.3.6. Tablet Mechanical Characteristics

A tableting study was undertaken on pure DAP raw material, SE co-crystals, co-crystal granules, and a physical mixture (PM) of a SE co-crystal with MCC and HPMC at the same compositions as the granule formulation. Since both solubility and dissolution are two critical parameters to be considered for designing an oral solid dosage form, the solubility of the co-crystals was considered before the tableting study. From Chappa et al.’s study [14], the higher the PEG molecular weight, the lower the solubility of the DAP-PEG co-crystals in various media (pH 1.2, pH 4.5, pH 6.8 buffers, and Milli Q water), so the DAP-PEG 4000 co-crystal and DAP-PEG 6000 co-crystal were not included in the tableting study. 

The tableting characteristics of DAP, SECC600A, SECC1000A, and SE1500A are shown in Figure 11(Ia,Ib). In terms of tabletability (tensile strength vs. compression pressure), tablets of SECC600A could not be prepared at a compression pressure ≤150 MPa, and a pressure of around 200 MPa was required for tableting. Although there was no tablet capping or lamination in the tablets of DAP prepared at a compression pressure between 80 MPa and 120 MPa, the tablets were so fragile that minimal horizontal pressure provided from the tablet hardness instrument led to tablet fragmentation. Nevertheless, SECC1000A and SECC1500A could dramatically enhance the tabletability of DAP resulting in higher ejection forces. This suggests that co-crystals comprised of higher molecular weight PEG coformers have better tableting properties. 

The tableting study of F1, F5, and the PM of the same composition as F5 is shown in Figure 11(IIa,IIb). Tabletability of DAP-PEG 600 co-crystals was improved by granulation (F5) and by physical mixing with MCC. Nonetheless, F5 has not only a greater tabletability over its PM, but also has an identical ejection force compared to its PM. This is consistent with the previous discussion, where FBG is capable of producing granules with high porosity that will improve tabletability [57]. Moreover, F1 has an improved tabletability compared to F5. However, it should be noted that F1 and F5 have different drug loadings. The improved tabletability of F1 might be caused by the higher MCC proportion in its formulation compared to F5 (Table 3 and Table 11). 

Given that granules F1 and F5 performed well in the preliminary tabletability studies presented here, future work will make use of a rotary tablet press to comprehensively investigate the tabletability, friability, and disintegration of tablets of F1 and F5 to more fully understand the potential industrial application of polymeric co-crystals.

#### 3.3.7. Tablet Dissolution Characteristics

The dissolution study of tablets prepared with the F5 granule formulation and its equivalent PM (prepared from SECC, MCC, and HPMC) was performed according to the BP method [31]. F5 was selected as Chappa et al. determined that only DAP-PEG 600 co-crystal showed a higher solubility compared to DAP across the physiological pH conditions, while co-crystals prepared with higher molecular weight PEG displayed more than 2.5 times lower solubility than DAP. 

Both tablets (F5 and its equivalent PM, the properties of the tablets are shown in Figure 11(IIa,IIb)) immediately dissolved in 0.1 M HCl after submersion (Figure 12). The tablets from the PM attained 71.3 ± 1.3% release in the first 5 min, 83.5 ± 1.0% release in 10 min, and leveled out at approximately 98% released after 20 min. F5 tablets had a much slower release than the PM of F5 over the first 10 min with 30.4 ± 1.1% release in 5 min and 56.0 ± 3.7% release in 10 min, but it still fully released at 30 min and there was no precipitation of DAP observed during the dissolution study. The reason why F5 tablets dissolved more slowly than F5 PM tablets might be that the F5 PM tablets disintegrated faster than F5 tablets (determined visually), because of their lower tensile strength when compressed at 200 MPa. Nevertheless, F5 tablets with a higher tensile strength still achieved more than 70% release at 45 min, as is required by the British Pharmacopoeia [58].

## 4. Conclusions

In this study, we have successfully proven the possibility of one step in situ co-crystallization and granulation for polymer co-crystals by FBG. Furthermore, FBG has been proven to be more efficient in producing polymer co-crystals than the traditional SE method, since the polymer co-crystals produced by FBG have a higher purity than the polymer co-crystals recovered from SE. Moreover, the DAP-PEG 400 co-crystal, which was not previously reported, has shown its possibility of acting as a binder in FBG. This might expand the application of polymer co-crystals, which is worth consideration in the co-crystal screening phase. All granules produced have shown improved flow characteristics, tableting properties, and dissolution characteristics compared to DAP, DAP-PEGs co-crystals on their own, and a PM containing a DAP-PEG co-crystal. In conclusion, FBG is a suitable processing technique for the production of granule formulations of polymer co-crystals. Further studies involving other APIs capable of forming such polymer co-crystals are warranted. 

## Figures and Tables

**Figure 1 pharmaceutics-15-02330-f001:**
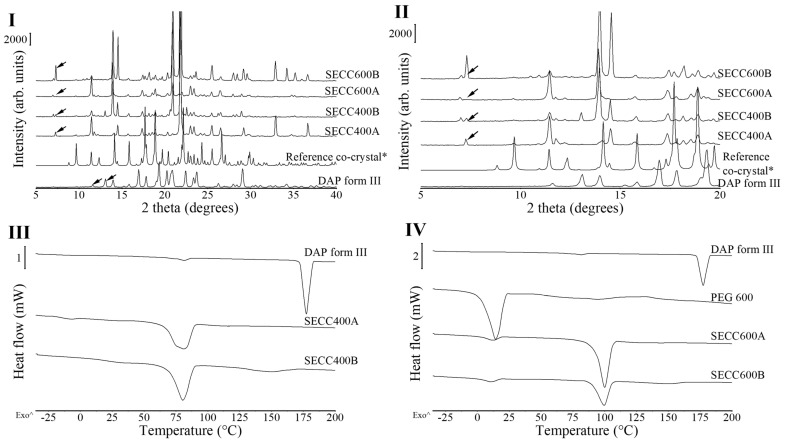
PXRD patterns and DSC thermograms of DAP and PEG 600 raw materials, SECC400A/B and SECC600A/B. *: simulated PXRD pattern of DAP-PEG 600 co-crystal with CCDC identifier number of 1868167 [14]. (**I** and **II**) PXRD diffractograms of SECC400A/B and SECC600A/B. (**III**) DSC thermograms of SECC400A/B. (**IV**) DSC thermograms of SECC600A/B. Arrows indicate DAP polymorphs. The arrow (^) indicates the direction of exothermic events.

**Figure 2 pharmaceutics-15-02330-f002:**
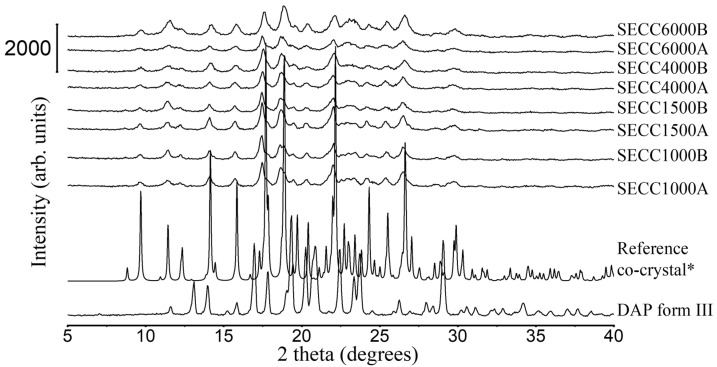
PXRD patterns of DAP raw material, SECC1000A/B, SECC1500A/B, SECC4000A/B, and SECC6000A/B. *: simulated PXRD pattern of DAP-PEG 600 co-crystal with CCDC identifier number of 1868167 [14].

**Figure 3 pharmaceutics-15-02330-f003:**
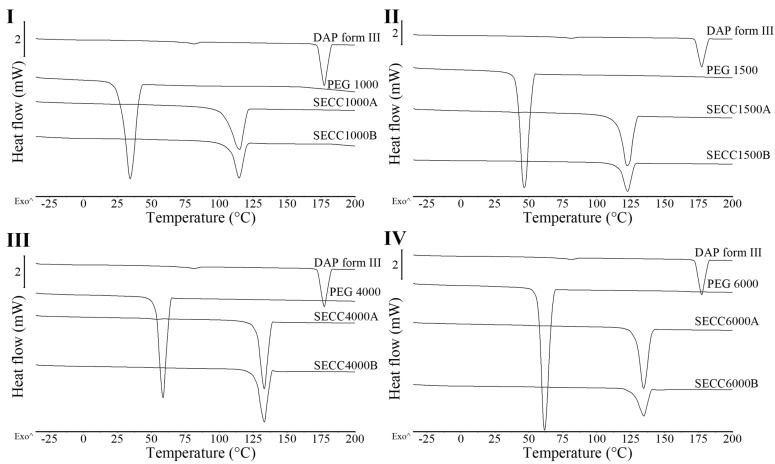
DSC thermograms of DAP, PEG 1000, PEG 1500, PEG 4000, and PEG 6000 raw materials, SECC1000A/B, SECC1500A/B, SECC4000A/B, and SECC6000A/B. (**I**) DSC thermograms of SECC1000A/B, (**II**) DSC thermograms of SECC1500A/B, (**III**) DSC thermograms of SECC4000A/B, and (**IV**) DSC thermograms of SECC6000A/B. The arrow (^) indicates the direction of exothermic events.

**Figure 4 pharmaceutics-15-02330-f004:**
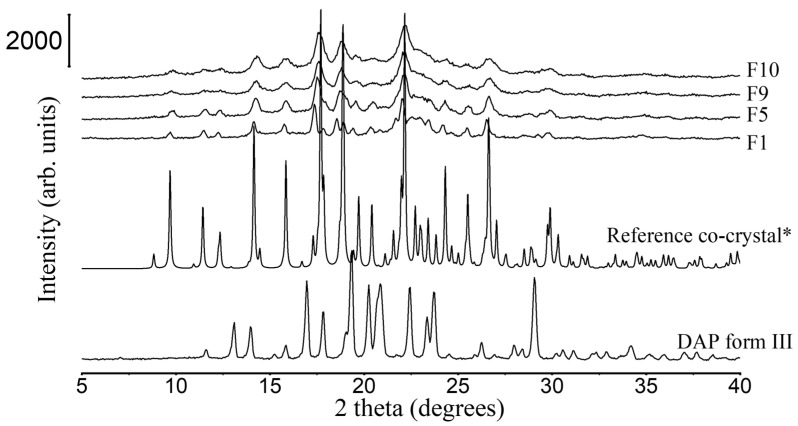
PXRD patterns of DAP raw material, F1, F5, F9, and F10 granules. *: simulated PXRD pattern of DAP-PEG 600 co-crystal with CCDC identifier number of 1868167 [14].

**Figure 5 pharmaceutics-15-02330-f005:**
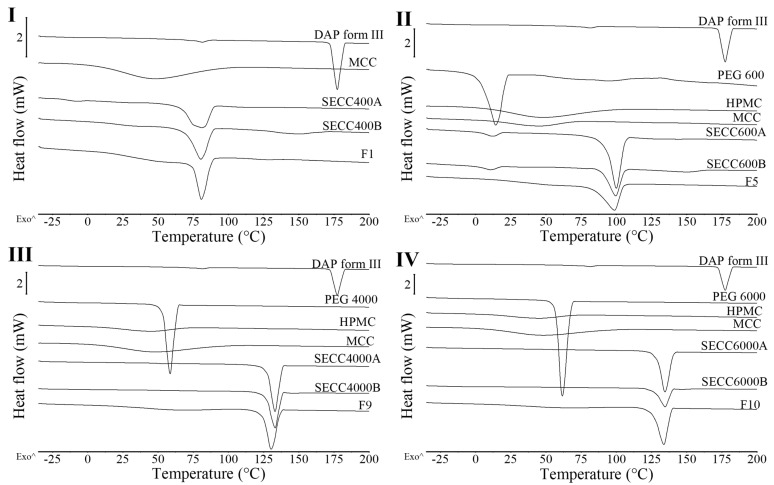
DSC thermograms of DAP, MCC, HPMC, PEG 600, PEG 4000, PEG 6000 raw materials, F1, F5, F9, and F10 granules. (**I**) DSC thermograms of F1 and SECC400A/B. (**II**) DSC thermograms of F5 and SECC600A/B. (**III**) DSC thermograms of F9 and SECC4000A/B. (**IV**) DSC thermograms of F10 and SECC6000A/B. The arrow (^) indicates the direction of exothermic events.

**Figure 6 pharmaceutics-15-02330-f006:**
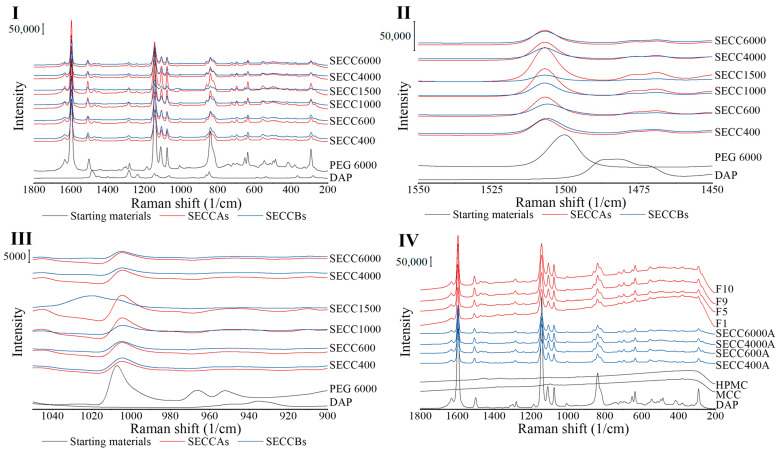
Raman spectra of DAP, PEG 6000, MCC, HPMC raw materials, SECC400A/B, SECC600A/B, SECC1000A/B, SECC1500A/B, SECC4000A/B, SECC6000A/B, F1, F5, F9, and F10 granules. (**I**, **II**, and **III**) Raman spectra of SECC400A/B, SECC600A/B, SECC1000A/B, SECC1500A/B, SECC4000A/B, SECC6000A/B. (**IV**) Raman spectra of SECC400A, SECC600A, SECC4000A, SECC6000A, F1, F5, F9 and F10.

**Figure 7 pharmaceutics-15-02330-f007:**
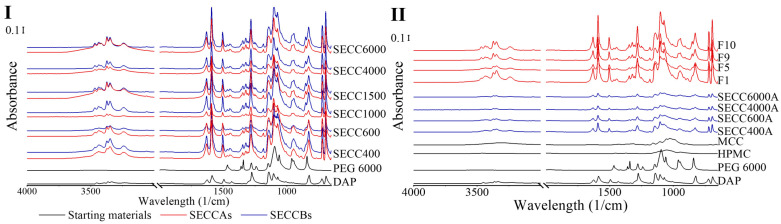
Raman spectra of DAP, PEG 6000, HPMC, MCC raw materials, SECC400A/B, SECC600A/B, SECC1000A/B, SECC1500A/B, SECC4000A/B, SECC6000A/B, F1, F5, F9, and F10 granules. (**I**): FTIR spectra of SECC400A/B, SECC600A/B, SECC1000A/B, SECC1500A/B, SECC4000A/B, SECC6000A/B. (**II**): FTIR spectra of SECC400A, SECC600A, SECC4000A, SECC6000A, F1, F5, F9 and F10.

**Figure 8 pharmaceutics-15-02330-f008:**
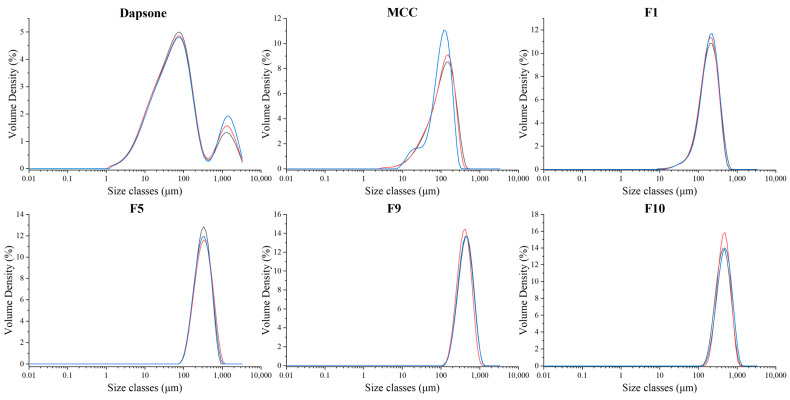
Particle size distributions of DAP raw material, MCC raw material, F1, F5, F9, and F10 granules at 2 bar (n = 3). Each line represents one analysis.

**Figure 9 pharmaceutics-15-02330-f009:**
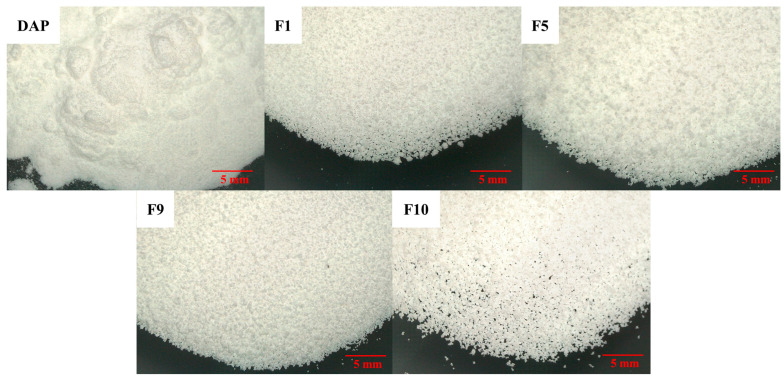
Light microscopy of DAP raw material, F1, F5, F9, and F10 granules.

**Figure 10 pharmaceutics-15-02330-f010:**
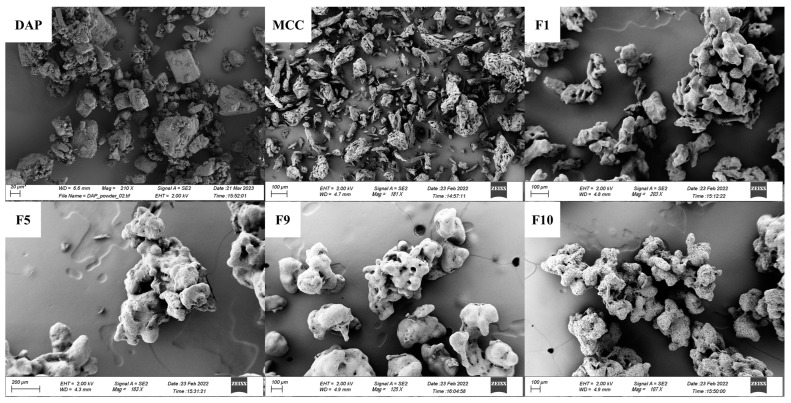
Particle morphology study by SEM on DAP raw material, MCC raw material, F1, F5, F9, and F10 granules.

**Figure 11 pharmaceutics-15-02330-f011:**
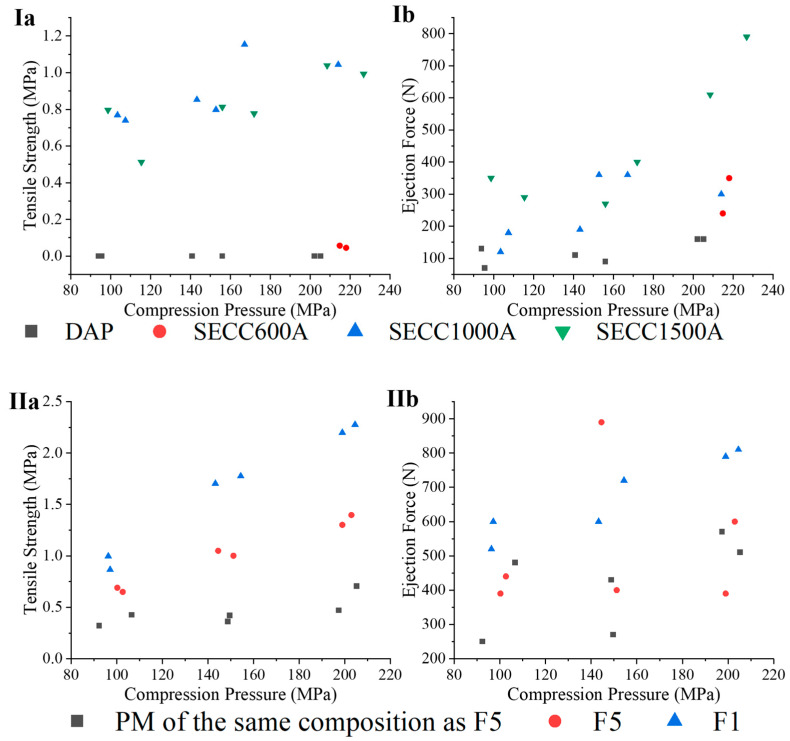
(**Ia**,**Ib**) tensile strength and ejection force for DAP, SECC600A, SECC1000A, SECC1500A. (**IIa**,**IIb**) tensile strength and ejection force for F1, F5, and PM of the same composition as F5.

**Figure 12 pharmaceutics-15-02330-f012:**
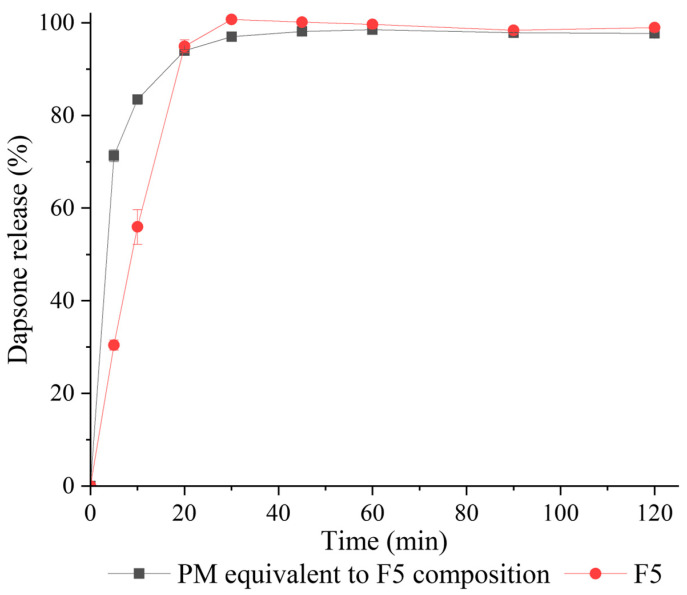
Dissolution profile of tablets prepared from F5 granules (red circles) and a PM equivalent to F5 composition (black squares) in 0.1 M HCl.

**Table 1 pharmaceutics-15-02330-t001:** Co-crystals produced by solvent evaporation (SE); PEG molecular weight (MW) used and weight ratio of DAP:PEG.

Abbreviation	PEG MW Used	DAP:PEG Ratio (*w/w*)
SECC400A	400	55.0:45.0
SECC400B	400	57.5:42.5
SECC600A	600	55.0:45.0
SECC600B	600	57.5:42.5
SECC1000A	1000	55.0:45.0
SECC1000B	1000	57.5:42.5
SECC1500A	1500	55.0:45.0
SECC1500B	1500	57.5:42.5
SECC4000A	4000	55.0:45.0
SECC4000B	4000	57.5:42.5
SECC6000A	6000	55.0:45.0
SECC6000B	6000	57.5:42.5

**Table 2 pharmaceutics-15-02330-t002:** Fluidized bed granulation (FBG) process variables.

FBG Process Variables	Airflow Rate (m^3^/h)	Temperature (°C)	Atomization (bar)	Liquid Feed Rate (mL/min)	Blowback Cycle
P1	20	30	0.5	7.5	10
P2	20	40	0.5	3.0	10

**Table 3 pharmaceutics-15-02330-t003:** Fluidized bed granulation (FBG) granule formulations’ compositions.

Formulation Code	PEG MW	[Filler (MCC) + Binder (HPMC)]: (DAP+PEG) (*w*/*w*)	Solvent Volume (mL)	FBG Process Variables	Granulation Outcome
F1	PEG 400	[70 + 0]:(30)	30	P1	Y
F2	PEG 600	[70 + 0]:(30)	200	P1	N
F3	PEG 400	[70 + 0]:(30)	30	P2	N
F4	PEG 400	[65 + 5]:(30)	100	P2	N
F5	PEG 600	[65 + 5]:(30)	200	P2	Y
F6	PEG 600	[65 + 5]:(30)	500	P2	N
F7	PEG 1000	[65 + 5]:(30)	200	P2	N
F8	PEG 1500	[65 + 5]:(30)	200	P2	N
F9	PEG 4000	[65 + 5]:(30)	500	P2	Y
F10	PEG 6000	[65 + 5]:(30)	1000	P2	Y
**Formulation Code**	**PEG MW**	**[Filler (MCC) + Binder (HPMC)]:** **(DAP) (*w/w*)**	**Solvent Volume (mL)**	**FBG Process Variables**	**Granulation Outcome**
F11	N/A *	[65 + 5]:(30)	100	P2	N
F12	N/A *	[65 + 5]:(30)	200	P2	N
F13	N/A *	[65 + 5]:(30)	500	P2	N

* N/A: not applicable. Y: granules were obtained. N: granules was not obtained.

**Table 4 pharmaceutics-15-02330-t004:** Summary of endothermic events observed for SECC400A, SECC400B, SECC600A, and SECC600B.

	Co-Crystal Endothermic Melting	PEG Endothermic Melting	3rd Endothermic Event
SECC400A	77.5 ± 2.9 °C (72.8 ± 5.4 J/g)	−9.0 ± 0.4 °C (2.1 ± 1.2 J/g)	/
SECC400B	74.6 ± 9.3 °C (45.8 ± 8.3 J/g)	/	151.4 ± 2.3 °C (9.4 ± 1.7 J/g)
SECC600A	99.5 ± 0.6 °C (86.4 ± 8.2 J/g)	10.6 ± 0.7 °C (6.9 ± 1.1 J/g)	136.6 °C * (3.1 J/g) *
SECC600B	99.6 ± 0.1 °C (69.2 ± 11.2 J/g)	10.2 ± 0.3 °C (9.7 ± 1.2 J/g)	153.4 ± 2.6 °C (12.0 ± 4.6 J/g)

* Endothermic event only observed once in triplicate mDSC study.

**Table 5 pharmaceutics-15-02330-t005:** Summary of endothermic events observed for SECC1000-6000.

SE Samples	Co-Crystal Endothermic Melting	PEG Endothermic Melting	3rd Endothermic Event
SECC1000A	114.7 ± 0.2 °C (114.3 ± 1.1 J/g)	/	/
SECC1000B	115.0 ± 0.7 °C (114.0 ± 1.6 J/g)	/	/
SECC1500A	122.8 ± 0.4 °C (125.5 ± 2.9 J/g)	41.5 ± 0.7 °C (0.8 ± 0.4 J/g)	/
SECC1500B	123.1 ± 0.6 °C (32.7 ± 3.6 J/g)	/	/
SECC4000A	133.0 ± 0.5 °C (137.7 ± 5.9 J/g)	53.9 ± 0.3 °C (1.2 ± 0.1 J/g)	/
SECC4000B	132.8 ± 0.5 °C (89.2 ± 9.1 J/g)	/	145.8 ± 0.3 °C (1.5 ± 0.3 J/g)
SECC6000A	134.8 ± 0.7 °C (133.1 ± 1.4 J/g)	/	/
SECC6000B	134.7 ± 0.3 °C (63.7 ± 2.5 J/g)	/	147.7 ± 1.0 °C (1.0 ± 0.3 J/g)

**Table 6 pharmaceutics-15-02330-t006:** Summary of endothermic events observed for granules.

Granules	Co-Crystal Endothermic Melting	PEG Endothermic Melting	3rd Endothermic Event
F1	81.5 ± 1.2 °C (46.1 ± 5.0 J/g)	/	/
F5	98.3 ± 0.8 °C (48.6 ± 4.1 J/g)	/	/
F9	130.2 ± 0.1 °C (72.4 ± 0.8 J/g)	/	/
F10	133.7 ± 0.1 °C (71.8 ± 0.9 J/g)	/	/

**Table 7 pharmaceutics-15-02330-t007:** Residual solvent content (RSC) for recovered granules F1, F5, F9, and F10.

Granules	RSC (%)
F1	1.7 ± 0.1
F5	1.0 ± 0.1
F9	1.1 ± 0.1
F10	1.2 ± 0.1

**Table 8 pharmaceutics-15-02330-t008:** The Raman peak shifting and their assignments [47,48], after co-crystallization, concluded from Figure 6.

Raman Shift (1/cm)	Assignment
1500 (DAP) → 1507 (co-crystals)	C-C stretch of DAP
1487 (PEG) → None	CH_2_–CH_2_ symmetric bending vibration
1008 (DAP) → 1005 (co-crystals)	CCC in-plane bend
966 and 952 (DAP) → None	γCH

**Table 9 pharmaceutics-15-02330-t009:** FTIR bond stretching frequencies in DAP and DAP-PEG co-crystals.

FTIR Bond Stretching Frequency (1/cm)	Assignment
3455 (DAP) → 3471 (co-crystals)	NH_2_ asymmetric stretching
3396, 3366, 3344 (DAP) → 3437, 3375, 3347 (co-crystals)	N-H stretching
3234 (DAP) → 3241 (co-crystals)	N-H symmetric and asymmetric stretching
1626 (DAP) → 1630 (co-crystals)	NH_2_ bending

**Table 10 pharmaceutics-15-02330-t010:** Drug loading of co-crystal granules.

Granules	DAP Loading (%)	Co-Crystal Loading (%)
F1	23.1 ± 0.5	42.0 ± 0.9
F5	28.1 ± 1.4	51.1 ± 2.6
F9	29.7 ± 0.9	53.9 ± 1.7
F10	32.5 ± 0.8	59.1 ± 1.4

**Table 11 pharmaceutics-15-02330-t011:** Particle size analysis of DAP, MCC, and DAP-PEGs co-crystals granules.

	d_10_ (μm)	d_50_ (μm)	d_90_ (μm)	Span
DAP	10.3 ± 0.3	60.6 ± 2.3	926.3 ± 215.1	12.1
DAP *	7.6 ± 0.2	41.6 ± 0.7	150.4 ± 2.7	3.4
MCC	30.4 ± 0.5	108.0 ± 4.2	218.3 ± 27.7	1.9
F1	84.1 ± 2.3	190.0 ± 3.8	350.5 ± 11.9	1.5
F5	166.7 ± 2.3	315.2 ± 7.4	568.0 ± 30.9	1.3
F9	238.7 ± 6.9	418.9 ± 20.2	701.9 ± 44.9	1.1
F10	262.2 ± 14.7	449.7 ± 19.4	736.7 ± 38.6	1.1

* Particle size analysis was undertaken at 3 bar pressure.

**Table 12 pharmaceutics-15-02330-t012:** Hausner ratio of DAP raw material, F1, F5, F9, and F10, and their flow description [56].

Granules	Hausner Ratio	Flow Description
DAP	1.37 ± 0.05	“Poor”
F1	1.15 ± 0.01	“Good”
F5	1.18 ± 0.01	“Good”
F9	1.08 ± 0.02	“Excellent”
F10	1.16 ± 0.03	“Good”

## Data Availability

Not applicable.

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
