# Peer review of "One Step In Situ Co-Crystallization of Dapsone and Polyethylene Glycols during Fluidized Bed Granulation"

_pharmaceutics, 2023, doi:10.3390/pharmaceutics15092330_

Round 1
Reviewer 1 Report
This manuscript describes an interesting work where a drug – PEG cocrystal system was prepared using a FBG process. The authors performed systematic work to show that the FBG process is superior to the solvent evaporation process in terms of phase purity of the cocrystals and its ability to produce processable granules for tableting.
L86, do you mean “obtained” by “recovered”?
Table 3, add a column “granulation outcome” here. Later (L393-394), whether or not granules form was referred to Table 6, which does not contain any information on granulation.
L234, add “(MPa)” after “Tensile strength”
L236-237 and equation 4: They appear to be referring to tablet friability. However, such information was not mentioned in the manuscript. Tablet friability information should be added to show tablets are of acceptable quality per the pharmacopoeial standards.
Figure legends are generally very difficult to read. Authors should improve readability of all figures. For examples by placing sample label next to the data trace/curve/pattern
Figure 1, indicate characteristic peaks of each form to aid the discussion in the text. The single intense peak in Figure 1I make it difficult to compare patterns. Try to replot data by not showing the full most intense peak.
L319-324, This part is unclear.
Figure 3, legend, L352, “SECC4000” to “SECC4000B”
L449, delete “accurate”
Figures 6 & 7: use darker colors for easier read.
Figure 8, axis labels and numbers are too small to read
L535, The authors attribute to the larger particles in the distribution to agglomeration of fine particles. If this is the case, the method used for particle size measurement is not suitable. The air pressure for dispersing particles should be higher.
Friability data is probably more important than tabletability data. They should be presented and discussed.
Reviewer 2 Report
In the manuscript of Shao et al., the authors have studied the co-crystallization of dapsone and polyethylene glycols during fluidized bed granulation. It is a well-planned work. The methodology is correct, and the results are interesting. Only three minor comments:
1.- Line 293 the word that is repeated.
2.- In the legend of figure 10 the authors should explain the reason for three results reported. Are the results of three different measurements of the same sample?
3.- In the methodology section there is a description of the stability study, and the results of stability are reported in the supplementary file but not in the results and discussion of the manuscript. The authors should include the comments about stability in the manuscript or remove it from the methodology and supplementary section.
Round 2
Reviewer 1 Report
Based on the explanation by the author, the values calculated using Eq. 4 are not for friability. Does “weight before” means the amount of powder filling the die? If so, this parameter does not have to do with friability. Instead, it was caused by particle escaping the die from the gap between the lower punch and the die bore during compaction. This value should be generally very small since the gap is normally very small. Suggest to delete the added discussion in L602-605, Eq. 4, and and any reference to it from the manuscript.
